# Exploring the Potential of *Lactobacillus helveticus* R0052 and *Bifidobacterium longum* R0175 as Promising Psychobiotics Using SHIME

**DOI:** 10.3390/nu15061521

**Published:** 2023-03-21

**Authors:** Fellipe Lopes De Oliveira, Mateus Kawata Salgaço, Marina Toscano de Oliveira, Victoria Mesa, Adilson Sartoratto, Antonio Medeiros Peregrino, Williams Santos Ramos, Katia Sivieri

**Affiliations:** 1Graduate Program in Food, Nutrition, and Food Engineering, Campus Araraquara, São Paulo State University (UNESP), Araraquara 14800-060, SP, Brazil; 2University of Araraquara—UNIARA, Araraquara 14801-320, SP, Brazil; 3Université Paris Cité, INSERM, UMR-S 1139 (3PHM), Faculty of Pharmacy, F-75006 Paris, France; 4Food and Human Nutrition Research Group, School of Nutrition and Dietetics, Universidad de Antioquia (UdeA), Medellín 050010, Antioquia, Colombia; 5CPQBA-UNICAMP, Paulínia 13148-218, SP, Brazil; 6Psychiatric Department, Pernambuco State University (UPE), Recife 50100-010, PE, Brazil; 7APSEN Farmacêutica, Department of Medical Affairs, Santo Amaro 04753-001, SP, Brazil

**Keywords:** psychobiotic, colonic fermentation, microbiota–brain-axis

## Abstract

Psychobiotics are probiotics that have the characteristics of modulating central nervous system (CNS) functions or reconciled actions by the gut–brain axis (GBA) through neural, humoral and metabolic pathways to improve gastrointestinal activity as well as anxiolytic and even antidepressant abilities. The aim of this work was to evaluate the effect of *Lactobacillus helveticus* R0052 and *Bifidobacterium longum* R0175 on the gut microbiota of mildly anxious adults using SHIME^®^. The protocol included a one-week control period and two weeks of treatment with *L. helveticus* R0052 and *B. longum* R0175. Ammonia (NH4+), short chain fatty acids (SCFAs), gamma-aminobutyric acid (GABA), cytokines and microbiota composition were determined. Probiotic strains decreased significantly throughout the gastric phase. The highest survival rates were exhibited by *L. helveticus* R0052 (81.58%; 77.22%) after the gastric and intestinal phase when compared to *B. longum* (68.80%; 64.64%). At the genus level, a taxonomic assignment performed in the ascending colon in the SHIME^®^ model showed that probiotics (7 and 14 days) significantly (*p* < 0.005) increased the abundance of *Lactobacillus* and *Olsenella* and significantly decreased *Lachnospira* and *Escheria-Shigella*. The probiotic treatment (7 and 14 days) decreased (*p* < 0.001) NH_4_^+^ production when compared to the control period. For SCFAs, we observed after probiotic treatment (14 days) an increase (*p* < 0.001) in acetic acid production and total SCFAs when compared to the control period. Probiotic treatment increased (*p* < 0.001) the secretion of anti-inflammatory (IL-6 and IL-10) and decreased (*p* < 0.001) pro-inflammatory cytokines (TNF-alpha) when compared to the control period. The gut–brain axis plays an important role in the gut microbiota, producing SCFAs and GABA, stimulating the production of anti-anxiety homeostasis. The signature of the microbiota in anxiety disorders provides a promising direction for the prevention of mental illness and opens a new perspective for using the psychobiotic as a main actor of therapeutic targets.

## 1. Introduction

Mental health is a growing concern worldwide, especially after the COVID-19 pandemic era, in which rates of common disorders such as anxiety have increased [1]. Anxiety disorders constitute a group of pervasive, persistent, disabling conditions, and are associated with a high onus of the disease, ranked among the top 25 leading causes of burden worldwide [2]. In 2019, 301 million people were living with an anxiety disorder, including 58 million children and adolescents [3].

Although adherence to psychotherapeutic and psychotropic treatments has increased over the past years [4], the current treatment modalities are still challenging in efficacy regarding remission, prognosis, and relapse prevention [5]. Available drugs that ameliorate mood and anxiety, such selective serotonin reuptake inhibitors (SSRI), selective serotonin and noradrenaline reuptake inhibitors (SNRI) and tricyclics (TCA) frequently have negative secondary effects, such as dizziness, weight gain, sexual impairment and gastrointestinal consequences. Benzodiazepines, largely used to treat anxiety as well, by its turn, can impair cognition, vigilance and may cause tolerance and dependence [6]. Furthermore, there is a massive variation in response to existing treatments, which are overall efficacious in less than half of the diagnosed patients [7,8]. Therefore, developing alternative treatments or adjunctive therapies is an urgent need in this scenario. One such promising area of research is the microbiota–gut–brain axis. The human gut microbiome is inhabited by 10^13^ to 10^14^ microorganisms (bacteria, viruses, archaea, and fungi), which outnumber the 20,000–25,000 human genes by 10 million genes in the entire human microbiota [9,10].

The gut microbiota plays an important role in distinct physiological processes, including multiple functions (e.g., by producing vitamins and neurotransmitters, contributing to food digestion, regulating neuronal feeding circuits, drug biotransformation, expression of genes and others) [11,12], as well as immune functions (e.g., providing defense against pathogenic strains) [13].

The gut microbiota and the brain communicate with each other in a bidirectional way via multiple mechanisms, including directly through the vagus nerve, via endocrine and via immune systems [14,15]. This reciprocal interaction between the brain and gut microbiota is known as the “gut–brain-axis”. Gut microbiota thereby have the potential to influence brain activity and mental health [16]. Recent scientific evidence suggests that brain functions including mood, cognitive function, and stress-associated anxiety or depression in humans may be related to the gut microbiome axis [17,18].

A range of high quality preclinical and clinical studies have shown that the disruption of the gut microbiota modulates stress reactivity [19,20,21] and is connected to mental health outcomes [22]. One of the first studies that showed the brain–microbiota-axis was carried out by Sudo et al. [23] using germ-free (GF) male mice that have a hyperreactive HPA axis, leading to increased concentrations of corticosterone and adrenocorticotropic hormones (ACTH) after a stressful stimulus. Another interesting study was performed with the transplantation of depressed patients to microbiota-depleted rats, which induced the development of some of the behavioral and physiological features of the depressive phenotype [24]. Thus, these findings suggest that the gut microbiota exerts an important influence on mental health, shaping host immunity and stress responses [25]. Hence, intervention in the microbiota may hold promise for the treatment of mental illnesses. In this way, in 2013, Dinan et al. [11] introduced a psychobiotics concept, defined as a probiotic yielding neurobehavioral or psychiatry benefits.

The possible mechanisms of psychobiotics in mental illness have been investigated. In preclinical studies, the psychobiotics are proved to increase the integrity of the intestinal epithelial barrier, reducing permeability, inhibiting endotoxemia and presenting anti-inflammatory signals to immune cells [25]. For example, pretreatment with *Lactobacillus helveticus* and *Bifidobacterium longum* restored the tight junction barrier integrity in mice facing water avoidance stress [26]. *Lactobacillus rhamnosus* significantly reduced corticosterone levels in mice subjected to stress-induced hypothermia attenuating stress response, demonstrating the effects on anxiety- and depressive-like behaviors [20]. *Bifidobacterium breve* reduced anxiety-like behavior during an elevated maze test in the same anxious mouse strain [27]. Beneficial effects of probiotics have also been seen in clinical trials. *L. rhamnosus* and *Lactobacillus reuteri* reduced small intestinal permeability in children with eczema [28]. Probiotic strains, particularly the *Bifidobacterium* genus and *Lactobaciliaceae* family, help relieve stress, anxiety, and depression symptoms [29,30,31]. In human volunteers, the administration of a probiotic formulation consisting of *L. helveticus* R0052 and *B. longum* R0175A significantly attenuated psychological distress and reduced anxiety-like behavior, respectively [32]. Furthermore, the potential of *L. rhamnosus* to alter the expression of central GABA receptors, mediating depression and anxiety-like behavior, has been demonstrated [6]. GABA is the predominant inhibitory neurotransmitter in the nervous system and plays important physiological roles in humans, mediating several immunological [33] as well as intestinal neurophysiological processes [34,35]. This neurotransmitter can be synthesized by the gut microbiota residents, such as the *Lactobacillaceae* family and *Bifidobacteria* genus [6,36].

A recent metanalysis showed that probiotic intervention can improve the emotional state of depressed patients [32]. According to guidelines from The World Federation of Societies of Biological Psychiatry (WFSBP) and the Canadian Network for Mood and Anxiety Treatments (CANMAT) Taskforce, probiotic strains (at doses of 1–10 billion units per day) are provisionally recommended for adjunctive use and weakly recommended for monotherapy use in major depressive disorders [37]. However, probiotics with neurological effects are not universal, and it depends on strain type (isolated or combined), the administration method, intervention time, and the subjects’ physiological status [38]. In this way, given this ability of a probiotic to function by conferring benefits to the host, especially concerning mental health, the knowledge of their mechanisms and applications is imperative. Therefore, this study aimed to evaluate the effect of *Lactobacillus helveticus* R0052 and *Bifidobacterium longum* R0175 on the gut microbiota composition of individuals with mild anxiety using the validated microbiome model (Simulator of Human Intestinal Microbial Ecosystem^®^ (Shime^®^)) and measurement of metabolites.

## 2. Materials and Methods

### 2.1. Dynamic Colonic Microbiome Model

The Simulator of the Human Intestinal Microbial Ecosystem (SHIME^®^) consists of a dynamic model of the human gastrointestinal tract connected to a software, composed of five connected reactors that represent the stomach (R1), small intestine (R2) and the ascending colon (R3), transverse colon (R4) and descending colon (R5), with their respective pH values, residence time, temperature, and volumetric capacity. In this experiment, R4 and R5 were transformed into R3 to conduct a triplicate study of the ascending colon with a pH range from 5.6 to 5.9 and retention time of 20 h (Figure 1). For the conditions of the stomach, the pH ranged from 2.3 to 2.5 and with a retention time of 2 h. To simulate the passage through the duodenum, we used 60 mL of artificial pancreatic juice (12.5 g/L NaHCO_3_, 3.6 g/L Oxgall, 0.9 g/L pancreatin, Sigma-Aldrich, St. Louis, MO, USA) at a flow rate of 4 mL/min and time retention of 4 h. The pH of the reactors was automatically adjusted with the addition of sodium hydroxide or hydrochloric acid and the whole simulator remained at 37 °C. The anaerobiosis conditions of the simulator were performed with a nitrogen injection of 30 min/day [39,40,41].

Initially, ascending colons were added to 500 mL of sterile feed medium (1.0 g/L arabinogalactan (Sigma), 2.0 g/L pectin (Sigma), 1.0 g/L xylan (Roth, Karlsruhe, Germany), 3.0 g/L starch (Êxodo, Hortolândia, Brazil), 0.4 g/L glucose (Synth, Diadema, Brazil), 3.0 g/L yeast extract (Neogen, Lansing, MI, USA), 1.0 g/L peptone (Kasvi, Italy), 4.0 g/L mucin (Sigma) and 0.5 g/L cysteine (Sigma)) along with 40 mL of fecal inoculum supernatant [43].

Inclusion criteria for donors were young people, ages 20 to 30 years old, with mild anxiety. The Hamilton Anxiety Rating Scale [44], Bristol stool form scale (BSFS) [45] and constipation assessment scale (CAS) were used by a nutritionist to select the fecal donors (Table 1). Exclusion criteria were use of antibiotics in the last 6 months, and prebiotics or probiotics, medication for gastrointestinal or metabolic disease, and dietary supplements in the last 3 months.

### 2.2. Experimental Protocol

The experiment was conducted for 5 weeks in SHIME^®^, which were divided into: 2 weeks of microbiota stabilization (300 mL of the feed medium once a day) [46], 1 week of control period (240 mL of the feed medium plus 60 mL of pancreatic juice once a day) and 2 weeks of probiotic treatment (240 mL of the feed medium, 60 mL of pancreatic juice plus 3 pills of the probiotics (*Lactobacillus helveticus* R0052, 3.0 × 10^9^ cfu.log/pill and *Bifidobacterium longum* R0175, 3.0 × 10^8^ cfu.log/pill, Lallemand Health Solutions Inc., Montreal, QC, Canada)). Samples were collected every 7 days after the start of the control period, stored at −20 °C, and the experiment was carried out in biological triplicate.

### 2.3. Metabolic Activity: Ammonia (NH4+), Short-Chain Fatty Acids (SCFAs) and Gamma-Aminobutyric Acid (GABA) Production

A specific ammonia ion electrode (Model 95–12, Orion) is used for ammonium ions quantification [47].

The methodology to quantify SCFAs (acetic, propionic and butyric acids) was previous described by Dostal et al. [48]. Briefly, 2 mL of colonic fermented samples were centrifuged (14,000 rpm for 5 min), and 1 mL of the supernatant were diluted 1:1 with MilliQ water, filtered in Millex^®^ (0.45 μm), and then injected into an Agilent gas chromatograph (model HP-6890, Santa Clara, CA, USA), equipped with an Agilent selective mass detector (model HP-5975) using a DB-WAX capillary column (60 m × 0.25 mm × 0.25 μm) under the following conditions: temperature of injector = 220 °C, column = 35 °C, 2 °C/min, 38 °C; 10 °C/min, 75 °C; 35 °C/min, 120 °C (1 min); 10 °C/min, 170 °C (2 min); 40 °C/min, 170 °C (2 min) and detector = 250 °C. Helium was used as a drag gas at a flow rate of 1 mL/min. Stock solution of acetic, propionic, and butyric acids were used to construct the analytical curves. The samples were analyzed in triplicate per ascending colon replica, before and after treatment. Data was expressed in mmol/g [48].

GABA production was measured by a colorimetric test using GABA as the standard. The reactions occurred with 40 µL of supernatant 108 µL of Tris/HCl buffer (160 mM (Sigma)), 6.6 µL of 2-mercaptoethanol (100 mM (Sigma)), 10 µL of α-cetoglutarato (100 mM (Sigma)), 10 µL of NADP (25 mM (Sigma)), 5.5 µL of ultrapurewater and 20 µL of GABase (Sigma). The absorbance was measured at 340 nm every 15 s for 5 min in the centering of plates with 96 cups and results were expressed as mM of the GABA [49].

### 2.4. Survival of Lactobacillus helveticus R0052 and Bifidobacterium longum R0175 under Simulated GI Conditions

To assess the survival of probiotics under simulated GI conditions, samples were collected from R1 and R2 during the treatment period. For lactobacilli, the microdilution technique was used [50], followed by plating on MRS agar (Merck^®^). For bifidobacteria, one mL of the serial dilution was plated in MRS agar supplemented with lithium chloride (2 g/L (Merck^®^, Darmstadt, Germany)) and sodium propionate (3 g/L (Merck^®^, Germany)). Both samples were diluted in sterile peptone water. After, plates were incubated in anaerobic conditions for 72 h at 37 °C [51].

### 2.5. Microbiological Analysis Employing 16S rRNA Gene Sequencing

The extraction of bacterial DNA was performed using the DNeasy^®^ PowerSoil^®^ Pro Kit (QIAGEN, Hilden, Germany) following the manufacturer’s instructions. The DNA samples were then immediately frozen at −80 °C until molecular analysis. The library was prepared using primers for the V3–V4 region of 16S rRNA (~470 bp, amplified with primers 341F-806R), and bacteria amplicons were sequenced by Illumina platform (Novaseq6000 PE 250). Primers’ sequence 341F 5′-CCTAYGGGRBGCASCAG-3′ and 806R 5′-GGACTACNNGGGTATCTAAT-3′ [52].

Sequence data were processed and analyzed with QIIME (Quantitative Insights Into Microbial Ecology, version 2022.2.0 (https://qiime2.org/ (accessed on 15 December 2022)). On average, a total of 181,443 raw reads per sample were sequenced. Initially, during the demultiplex and trimming steps, the low-quality readings were removed, such as reads up to Q30, reads with unsatisfactory length and chimeras were removed with QIIME [53]. After this process, the data set contained an average of 27,154 raw reads per sample. The clean reads were used in the definition of the ASV (Amplicon Sequence Variant). To measure the rates present in the samples, a predictor model of the V3 and V4 regions was used (SILVA 138.99% OTUs from 515F/806R region of sequences). The operational taxonomic units (OTUs) were grouped by cluster readings with 99% similarity. The taxonomy was assigned to OTUs using SILVA 138 reference database (https://www.arb-silva.de/). Heatmaps and barplots of relative abundance of OTUs were generated with Python (version 3.7) through codes developed by the company ByMyCell Inova Simples Ltd., Ribeirão Preto, Brazil.

The rarefaction curve was made using QIIME. Alpha diversity was calculated by applying different metrics. The Shannon, Simpson and Fisher indices representing the species diversity were calculated. The Chao1 and ACE indices representing species richness were calculated. To evaluate the similarity of microbiota from different groups, the beta diversity was reported using weighted and unweighted UniFrac distances. A Permutational multivariate analysis of variance (PERMANOVA) test was employed using Adonis test, to evaluate differences of beta diversity groups. Predictive functional profiling of bacteria communities was identified by Phylogenetic Investigation of Communities by Reconstruction of Unobserved States 2 (PICRUSt2, version 2.4.2). The bacterial OTUs exported from QIIME2 in the standard format are imported into PICRUSt2. Exploration analysis of genomic data was carried out in Python (version 3.7) [54].

### 2.6. Co-Culture of Caco-2 and THP 1 Cells

The immunomodulatory effects of probiotic treatment were realized using a co-culture model of Caco-2 cells (HTB-37; American Type Culture Collection, Rio de Janeiro Cell Bank, Duque de Caxias, Brazil) and THP1 cells (TIB-202, Rio de Janeiro Cell Bank, Brazil). This co-culture model of Caco-2/THP1 cells was previously described by Daguet et al. [55]. Briefly, Caco-2 cells at passage 37 were seeded into 24-well semipermeable inserts (0.4 m Thincerts, Greiner Bio-one, São Paulo, Brazil) at a density of 80,000 cells/insert. Caco-2 cell monolayers were cultured for 14 days, with three medium changes/week, until a functional monolayer of cells with a transepithelial electrical resistance (TEER) of more than 300 Ω.cm^2^ was achieved. Cells were maintained in Dulbecco’s Modified Eagle Medium (DMEM; Life Technologies, São Paulo, Brazil) and supplemented with 10 mM HEPES (Life Technologies, São Paulo, Brazil) and 10% (*v*/*v*) heat-inactivated Roswell Park Memorial Institute (RPMI) 1640 medium containing fetal bovine serum (FBS; Life Technologies). THP1 cells were maintained in 11 mM glucose and 2 mM glutamine and supplemented with 10 mM HEPES, 1 mM sodium pyruvate and 10% (*v*/*v*) heat-inactivated FBS. THP1 cells at passage 29 were seeded in 24-well plates at a density of 500,000 cells/well and treated with 50 ng/mL phorbol 12-myristate 13-acetate (PMA; Sigma-Aldrich) for 48 h [55,56]. Prior to co-culture, the TEER of Caco-2 monolayers was evaluated using a Millicell ERS-2 (Merck Millipore, São Paulo, Brazil) Volt-Ohm epithelial meter (0 h time). The TEER of an empty well was subtracted from all readings to account for the residual electrical resistance of the insert. Then, Caco-2 pellets were placed on top of PMA differentiated THP1 cells for further experiments as described previously [56,57].

The apical compartment (containing the Caco-2 cells) was filled with sterile SHIME^®^ colonic suspensions (membranem, 0.22 µm) in Caco-2 culture media. The basolateral compartment (containing THP1 cells) was filled with Caco-2 culture media only. Cells were also exposed to Caco-2 culture media alone in both chambers as a control. The cells were treated for 24 h, after which the TEER was measured (24 h time point). After subtracting the TEER of the empty insert, all 24 h values were normalized to their own 0 h value (to account for differences in TEER of the different inserts). Then, the basolateral supernatant was discarded and cells were stimulated basolaterally with Caco-2 media culture containing 100 ng/mL LPS (*Escherichia coli* K12, InvivoGen, San Diego, CA, USA). Cells were also stimulated with media without LPS as a control. After 6 h of LPS stimulation, basolateral supernatant was collected for the measurement of cytokines such as human IL-6, IL-10 and TNF-α (Tumor necrosis factor alpha) by ELISA (eBioscience, Vienna, Austria). All treatments were performed in triplicate [56].

### 2.7. Statistical Analysis

We used paired *t*-test, one-way analysis of variance (ANOVA), and Tukey’s test to analyze the results of metabolic activity (Ammonia (NH4+), short-chain fatty acids (SCFAs) and gamma-Aminobutyric Acid (GABA) production) and survival of *L. helveticus* R0052 and *B.* R0175 under simulated GI conditions, with *p* < 0.05 considered statistically significant. A statistical analysis was performed using GraphPad Prism software, version 8.0 (La Jolla, CA, USA). Next, 16S rRNA gene sequence analyses were performed in RStudio, version 3.2.4 [58] using the phyloseq package [59] to import sample data and calculate alpha and beta diversity metrics. The significance of non-parametric variables (alpha diversity) was determined using the non-parametric Wilcoxon test for two category comparisons or the Kruskal–Wallis test when comparing three or more categories. Principal coordinate plots were based on the PERMANOVA test to estimate *p*-values [60]. The *p*-values were adjusted for multiple comparisons using the FDR algorithm [61].

## 3. Results

### 3.1. Survival of Lactobacillus helveticus R0052 and Bifidobacterium longum R0175 under Simulated GI Conditions

We demonstrated the survival of *L. helveticus* R0052 and *B. longum* R0175 in vitro gastrointestinal conditions. The *L. helveticus* R0052 and B. longum R0175 probiotic strains decreased significantly over the course of the gastric phase. The highest survival rates were exhibited by *L. helveticus* R0052 (81.58%; 77.22%) after the gastric and intestinal phases when compared to *B. longum* (68.80%; 64.64%). However, both strains reached the large intestine at least >7 cfu/mL (Figure 2).

### 3.2. Microbiota Composition in Long-Term SHIME^®^ Run

It is possible to notice that a total of 4,205,262 million high-quality reads were achieved from the microbiota samples collected over the periods of control (*n* = 9) and treatments (*n* = 9). Then, 3,217,336 million sequences were generated after normalizing the data.

Figure 3 shows the prevalent phyla in the microbiota of individuals with mild anxiety before and after dosing 7 and 14 days of *L. helveticus* R0052 and *B. longum* R0175. The predominant phyla before the probiotic treatment were *Actinobacteria*, *Firmicutes*, *Bacteriodetes* and *Proteobacteria*. However, after the dosing of probiotic, the increase (*p* < 0.001) of *Firmicutes* was observed (Figure 3).

We did not observe significant differences in alpha diversity between the control period and the probiotic treatment (7 and 14 days). However, beta diversity showed a significant difference (*p* < 0.001) between the control period and the probiotic treatment (7 and 14 days). Therefore, a principal-coordinate analysis showed different clustering in the control period, 7 days and 14 days after the probiotic treatment (Figure 4).

At the genus level, the taxonomic assignment performed in ascending colon vessels in the SHIME^®^ model showed that probiotics (7 and 14 days) significantly enhanced (*p* < 0.005) the abundance of *Lactobacillus* and *Olsenella* and significantly decreased *Lachnospira* and Escheria-Shigella (Figure 5).

### 3.3. Metabolic Activity: Ammonia (NH4+), Short-Chain Fatty Acids (SCFA) and Gamma-Aminobutyric Acid (GABA) Production

Metabolic activity involving NH4+ and SCFAs production is shown in Figure 6. The probiotic treatment (7 and 14 days) decreased (*p* < 0.001) the NH4+ production when compared with the control period. For the SCFAs, we observed after treatment with probiotics (14 days) an increase (*p* < 0.001) of acetic acid and total SCFAs when compared with the control period. However, for butyric and propionic acids, no statistical difference was observed in relation to the control period. In addition, after 14 days of probiotic treatment, we observed an increase (*p* < 0.001) of GABA production when compared with the control period (Figure 7).

### 3.4. Potential Modulation of the Gut-Epithelial Function and Immunity

The values of transepithelial electrical resistance (TEER) for Caco-2/THP1 cells treated with colonic media with and without treatment without probiotics (control) were lower (184. 8 ± 19.8) than with probiotics (7 days: 208.56 ± 35.7 and 14 days: 205.55 ± 31.1), but without statistical significance (*p* = 0.06).

Figure 8 shows the concentrations of anti-inflammatory cytokines IL-6 and IL-10 and pro-inflammatory cytokine TNF-α following exposure of the cells to colonic media after dosing with probiotics for 7 and 14 days. The treatment with probiotics increased (*p* < 0.001) the secretion of the anti-inflammatory and decreased (*p* < 0.001) proinflammatory cytokines when compared to the control period.

## 4. Discussion

In this study, we showed, for the first time, using a dynamic microbiome model, the positive influence of *Lactobacillus helveticus* R0052 and *Bifidobacterium longum* R0175 on gut microbiota of individuals with mild anxiety. Therefore, the probiotic ability to survive in adverse conditions in the gastrointestinal tract proves fundamental for probiotics to exert their beneficial effects on the colon [62]. Both strains evaluated reached the large intestine at least >7 cfu/mL. The international recommendation is that probiotics should be present in supplements in an amount of 8–9 log cfu/g in the daily recommendation of the product [63], before ingestion, to ensure that a sufficient therapeutic minimum of 6–7 cfu/g can reach the colon [38].

In the last 10 years, with the development and improvement of molecular tools, many researchers have been associating changes in the composition of the microbiota to an increase in potentially pathogenic groups and a decrease in beneficial ones, with different types of intestinal diseases and extra intestinal diseases, such as autism, depression, Alzheimer, dementia, Parkinson and others [64,65,66]. The researchers seek to know a signature of the microbiota to establish personalized treatment strategies. In this way, the high abundance of the *Actinobacteria* phylum in the microbiota of individuals with mild anxiety before the intervention with probiotics called our attention. However, after 14 days of probiotic treatment, we observed a decreased (*p* < 0.001) of this phylum. Interestingly, Simpson et al. [67], in a systematic review, showed that the increase of *Actinobacteria* in patients with anxiety positively correlated with depression symptoms. However, understanding and characterizing the gut microbiota involves a multiple approach, often including a measure of alpha and beta diversity [68]. The alpha diversity index summarizes the structure of an ecological community with respect to its richness (number of taxonomic groups), evenness (distribution of abundances of the groups), or both [69]. We observe a decrease in alpha diversity, but without statistical differences, after probiotic doses when compared with the control period. However, many studies that investigate the gut microbiota of patients with anxiety symptoms did not observe an association between the alpha diversity index and anxiety symptoms [70,71,72]. Conversely, after probiotic treatment, we observed a significant difference between the control period and the probiotic treatment in beta diversity, showing the influence of probiotics on microbiota composition. Normally, beta-diversity is measured by end-to-end comparison of microbiome pairs using distance metrics such as UniFrac or Bray–Curtis. Therefore, beta-diversity-based status identification and classification relies on an assumption that most members of the community, or at least the highly abundant members, are associated with the status of interest [73].

At genus level the increase of *Olsenella* is remarkable because this genus has been previously associated with a reduction in gut inflammation [74]. On the other hand, the increase of *Lactobacillus* spp. was expected, due to the high survival rate observed after gastric and duodenal digestion. Conversely, the decrease of *Lachnospira* and *Escherichia-Shigella* was an important result, since *Escherichia-Shigella* is known to be an opportunistic pathogen for humans, being able to produce several pro-inflammatory components, such as lipopolysaccharides and peptidoglycans, and being able to trigger the host immune response and lead to intestinal inflammation in varying degrees. In addition, Chen et al. [75] correlated a high abundance *of Escherichia/Shigella* with anxiety symptoms.

The gut microbiota composition and microbial metabolites have been associated with human health [76]. Hyperammoniac patients have a motor and cognitive dysfunction, indicating that the brain function can be affected by NH_4_ through underlying mechanisms [77]. Therefore, excessive production of ammonia by gut bacteria could contribute to increased ammonia levels in the blood and astrocyte edema [78]. We observed a strong decrease of NH_4_ after the probiotic treatment (7 and 14 days). Interestingly, Kurtz et al. [79] showed that engineered probiotic *Escherichia coli* Nissle 1917, which can convert NH_3_ to l-arginine, reduced systemic hyperammonemia in the preclinical model. However, the most crucial metabolic activity of gut bacteria is the production of SCFAs formed by the fermentation of gut bacteria [80]. The main SCFAs are acetic, propionic, and butyric acids. They contain one to six saturated carbons in their structures [81]. After the probiotics treatment, an increase of acetic acid was observed. The acetic acid has been reported to decrease a stimulate leptin, inflammatory cytokines and free fat acids to the liver [82]. In addition, SCFAs can enter the circulatory system; thus, this could probably be a signaling pathway between the microbiota and the host brain [83]. In this way, Zheng et al. [84] demonstrated that bacterial acetate production increased synaptophysin (SYP) in the hippocampus as well as learning and memory impairments level, improving cognitive decline in mice. In addition, Burton et al. [85] showed in clinical trials that SCFA increases are correlated with decreases of depressive and anxiety symptoms.

We observed in our study after 14 days of probiotic treatment an increase of gamma-aminobutyric acid (GABA). GABA is an important and well-known neurotransmitter in the central nervous system (CNS). Some species of *Lactobacillaceae* family and *Bifidobacterium* genus can produce GABA, among them we can highlight *L. helveticus* [12] and *B. longum* [86]. Additionally, Chen et al. [75] showed that *L. helveticus* upregulated the expression of GABA receptors in animal models. Interestingly, GABA serves as an acid resistance mechanism for some *Lactobacillaceae* and *Bifidobacterium* species. At a low pH (<5.0), glutamate decarboxylation is induced and produces GABA, which is then exported from the cell in a protonated form, alkalizing the cytoplasm [87]. In addition, the influence of the ingestion of some strains of probiotics and the improvement of behavior has been described. Patrick et al. [88] demonstrated that the probiotic *L. helveticus* R0052 and *B. longum* R0175 reduces the anxiety-like behavior in Syrian hamsters. In clinical trials, strains of *L. helveticus* and *B. longum* have been reported to improve psychological symptoms [89]. Probably, the outcomes observed in clinical trials with the strains *L. helveticus* R0052 and *B. longum* R0175 may be correlated with the GABA production observed in this study. Interestingly, a clinical trial found that the transplantation of a fecal microbiota from lean to obese individuals showed increased plasma GABA levels, showing that manipulating the microbiome can alter GABA levels [90].

Studies have shown that the gut microbiota can exert an effect on the hypothalamic– pituitary–adrenal axis [91,92]. Anxiety and depression disorders are correlated with dysregulation of the HPA axis, which is normally associated with higher levels of cortisol and inflammation [93]. The composition of the gut microbiota, mainly in dysbiosis, can contribute to the increase of cortisol and inflammation [94], but this effect is bilateral since pro-inflammatory states can aggravate the changes in microbiota composition through deleterious effects on gastrointestinal health [95]. Thus, excessive levels of circulating cortisol and inflammatory mediators increase intestinal permeability, allowing Gram-negative bacteria, which have lipopolysaccharides (LPS) in their membranes, to induce inflammation in the intestinal mucosa, thus being a trigger for a leaky gut. LPS can also translocate into the bloodstream, inducing chronic CNS inflammation [96]. We observed in our study the increase of anti-inflammatory cytokines (IL-6 and IL-10) and the decrease of TNF-α (pro-inflammatory) and a strong reduction of pro-inflammatory bacteria *Escherichia-Shigella* after 14 days of treatment with probiotics. IL-6 is known to associate with and aggravate an inflammation by differentiation directly/indirectly through the nervous system and lymphocytes proliferation [97]. However, the anti-inflammatory functions of IL-6 have been revealed. In this way, we highlight that IL-6 can help maintain mucosal integrity and facilitate intestinal barriers repair with an increased claudin-2 expression [98]. Interestingly, we observed an improvement trend in the transepithelial electrical resistance of Caco-2/THP1 cocultures, suggesting an increase in mucosal integrity.

## 5. Conclusions

The results found in SHIME showed that the combination of *L. helveticus* R0052 and *B. longum* R0175 can modulate the gut microbiota, increasing gamma-aminobutyric acid (GABA) and short chain fatty acids, decreasing pro-inflammatory cytokines and increasing the anti-inflammatory one. Therefore, *L. helveticus* R0052 and *B. longum* R0175 showed it is a promising psychobiotic for the adjuvant treatment of patients with anxiety. Finally, the microbiota signature in anxiety disorders provides a hopeful direction for the prevention of mental diseases and opens a new perspective of the use of psychobiotics as a main player of therapeutic targets.

## 6. Commentary of Expert: Dr. Antonio Medeiros Peregrino (Psychiatrist, Master in Neuropsychiatry and Behavioral Sciences and PhD in Tropical Medicine)

In recent decades, psychiatry has developed numerous lines of research for a better understanding of the etiopathogenesis and pathophysiology of anxious and depressive states. The aim is to achieve constant improvements in pharmacological therapy, in biological or physical interventions (e.g., electroconvulsive therapy, transcranial magnetic stimulation) and even in psychotherapies. As it has been observed that up to one third of depressed patients may not respond adequately to conventional pharmacological treatments, it is possible to assume that different pathophysiological mechanisms are involved in the genesis of depression. The same can be supposed for anxious pictures. The finding that intestinal dysbiosis may constitute one of the etiopathogenic factors in anxiety-depressive conditions directs psychiatry to pay attention to new areas of clinical activity. The use of psychobiotics (probiotics with neuropsychiatric action), especially *L. helveticus* and *B. longum*, has been an area of interest in the search for better monoamine homeostasis and, likewise, for the reduction of inflammatory processes underlying the disorders. Using the SHIME^®^ model, we were able to verify the increase of the gamma-aminobutyric acid (GABA) and of the short chain fatty acids, the reduction of pro-inflammatory cytokines and the increase of the anti-inflammatory ones, among other findings, indicating that knowledge of the brain–gut axis may be increasingly important in our daily clinical practice. The use of psychobiotics can thus be an important element, probably adjunctive, in the treatment of anxiety-depressive disorders.

## Figures and Tables

**Figure 1 nutrients-15-01521-f001:**
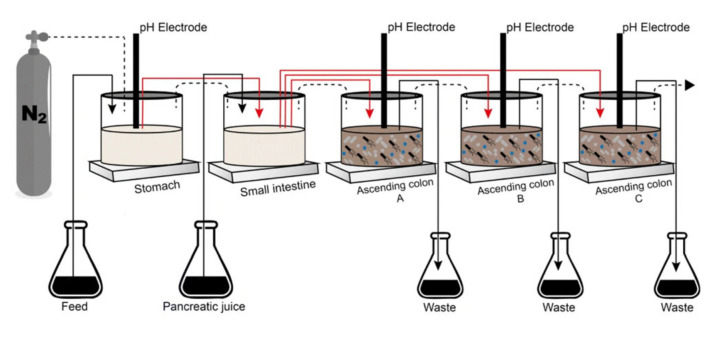
Simulator of the Human Intestinal Microbial Ecosystem (SHIME^®^) adapted adapted with permission from Ribeiro et al. [42].

**Figure 2 nutrients-15-01521-f002:**
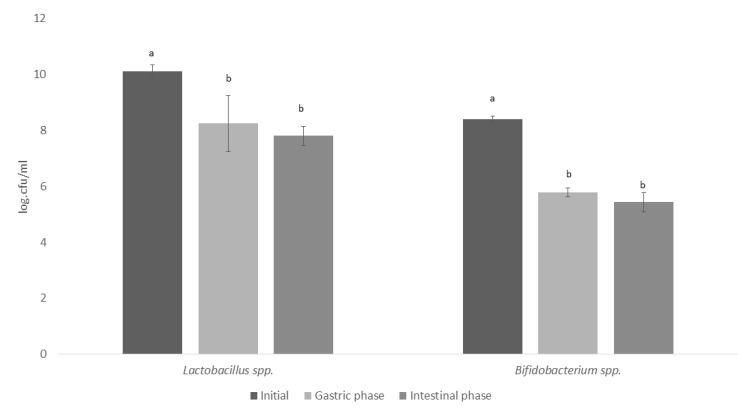
Survival of *Lactobacillus helveticus* R0052 and *Bifidobacterium longum* R0175 under simulated GI conditions in Simulator of the Human Intestinal Microbial Ecosystem (SHIME^®^). a,b, different superscript lowercase letters represent differences (*p* < 0.05), based on Tukey's test.

**Figure 3 nutrients-15-01521-f003:**
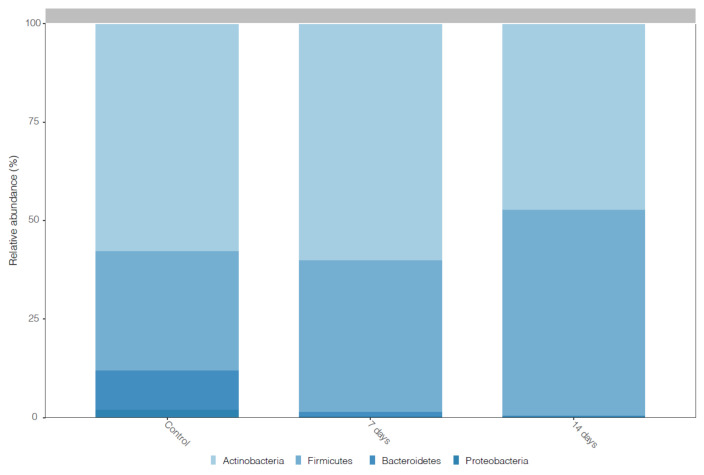
Histogram of the community composition of gut microbiota at the phylum level. Control = control period, 7 days: 7 days of treatment with *L. helveticus* R0052 and *B. longum* R0175 and 14 days: 14 days of treatment with *L. helveticus* R0052 and *B. longum* R0175.

**Figure 4 nutrients-15-01521-f004:**
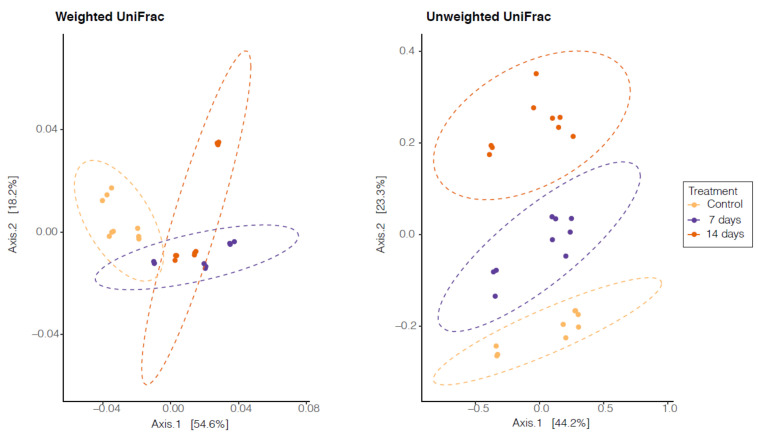
Principal Coordinate Analysis (PCoA) based on the abundance of operational taxonomic units (OTUs). Unweighted Unifrac *p* < 0.0002 and Weighted Unifrac *p* < 0.0001. Control = control period, 7 days: 7 days of treatment with *L. helveticus* R0052 and *B. longum* R0175 and 14 days: 14 days of treatment with *L. helveticus* R0052 and *B. longum* R0175.

**Figure 5 nutrients-15-01521-f005:**
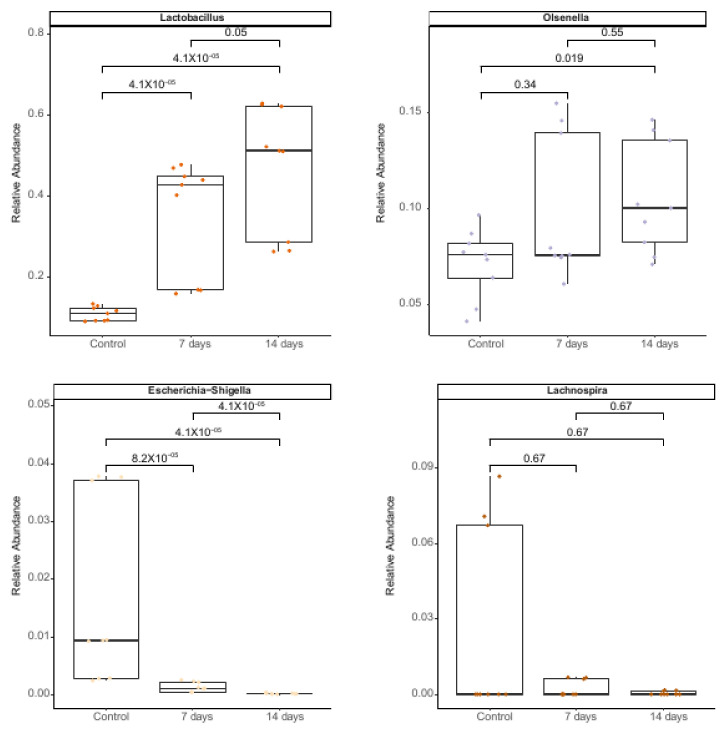
Relative abundance of bacterial genera in microbiota of individuals with mild anxiety using the SHIME^®^ model over the course of the 7 or 14 days of treatment with *L. helveticus* R0052 and *B. longum* R0175. Differences in means were analyzed using the Wilcoxon test.

**Figure 6 nutrients-15-01521-f006:**
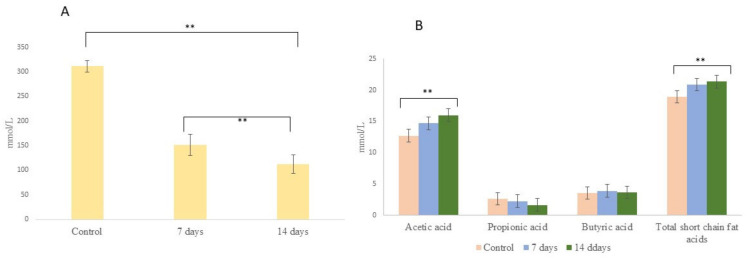
Ammonium (**A**) and short-chain fatty acid (**B**) concentrations in the ascending colon in microbiota of individuals with mild anxiety using the SHIME^®^ model over the course of the 7 or 14 days of treatment with *L. helveticus* R0052 and *B. longum* R0175. Asterisks (**) represent significant differences relative to control according to the paired *t*-test (** *p* < 0.01).

**Figure 7 nutrients-15-01521-f007:**
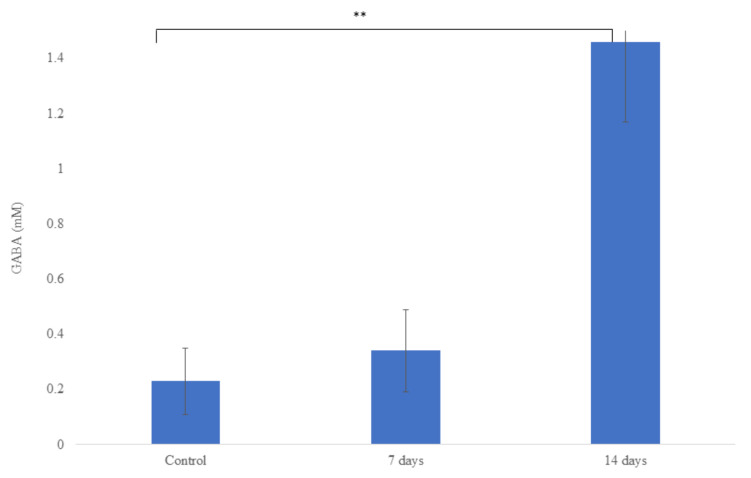
Gamma-Aminobutyric Acid (GABA) production in the ascending colon in microbiota of individuals with mild anxiety using the SHIME^®^ model over the course of the 7 or 14 days of treatment with *L. helveticus* R0052 and *B. longum* R0175. Asterisks (**) represent significant differences relative to control according to the paired *t*-test (** *p* < 0.01).

**Figure 8 nutrients-15-01521-f008:**
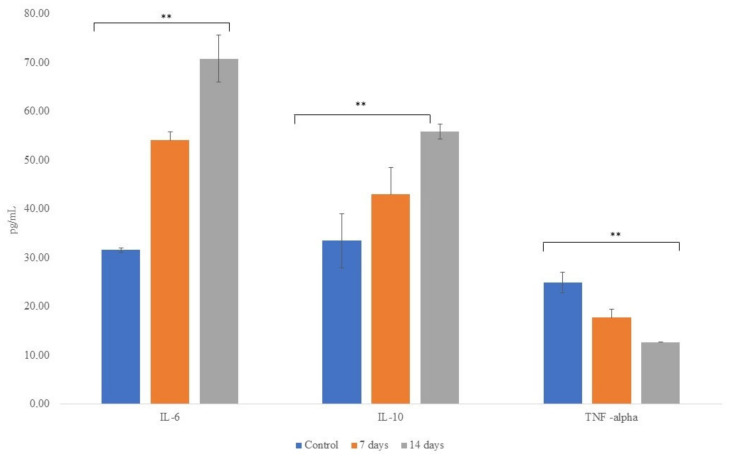
Concentrations of anti-inflammatory cytokines IL-6 and IL-10 and pro-inflammatory cytokine TNF-α following exposure of the cells to colonic media after dosing with *L. helveticus* R0052 and *B. longum* R0175 for 7 and 14 days. Asterisks (**) represent significant differences relative to control according to the paired *t*-test (** *p* < 0.01).

**Table 1 nutrients-15-01521-t001:** Characteristics of fecal pool donors.

Parameters	N = 3 (2 Women and 1 Men)
Hamilton Anxiety Rating Scale	13.3 ± 1.70 (mild anxiety)
Bristol stool form scale	mild constipation
Constipation assessment scale (CAS)	4.6 ± 0.94: (mild constipation)
Age (years)	29.3 ± 4.78
BMI (Body Mass Index)	22.1 ± 3.33

## Data Availability

All data generated by the current project are available upon request.

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
