# Peer review of "Exploring the Potential of Lactobacillus helveticus R0052 and Bifidobacterium longum R0175 as Promising Psychobiotics Using SHIME"

_nutrients, 2023, doi:10.3390/nu15061521_

Round 1

Reviewer 1 Report

I am wondering what was time between samples collection and 16S rRNA analysis. I am not sure that keeping samples at -20o C is safe. Usually fecal samples are kept at -80o C. This question should be answered in the manuscript

There is no control group of donors without anxiety in the study, and this significantly decrease the value of the manuscript. 

Author Response

We very much appreciate your review. Regarding the temperature, it was a typing error, as the samples were stored at -80C.

Reviewer 2 Report

This is a highly interesting study using a novel method (SHIME) to the utility of implementing psychobiotic in clinical practice. 

Despite the overall high merit of the study, the Authors should review the manuscript carefully and make many changes to improve the quality.

Firstly, a large number of typos should be corrected (for example, "citokines", "bad mental health outcomes", capitalization: "cfu or CFU" etc.)

Authors mentioned IL6 as an anti-inflammatory cytokine. However, most studies concerning psychiatric patients highlighted that this factor is engaged in the inflammatory response. Could Authors expand the information about IL6 in the Discussion section and raise the issue of an inflammatory properties?

The Introduction should be a bit shorter (compare the Introduction to the Discussion). Please, omit the least important information.

In the above-mentioned section Authors wrote about aggravating cognitive symptoms by antidepressants. Add information about class of drugs (SSRI, SNRI, benzodiazepines, TCA or other?)

"However, the probiotic effect is not universal (...)" Authors should cite guidelines from The World Federation of Societies of Biological Psychiatry (WFSBP) and the Canadian Network for Mood and Anxiety Treatments (CANMAT) Taskforce. According to the WFSBP, the antidepressant effect of probiotics is not strain-dependent. Please, discuss it. https://www.tandfonline.com/doi/full/10.1080/15622975.2021.2013041

In the Method section, the references of the scales should be added, and the information about the researcher who performed the clinical assessment.

In Figure 5, the Authors should edit the p-value (f.e., 4. te-05???).

Despite no TEER (p-value) changes in point 3.4, the p-value may be close to the significance. Please, add the p-value to this section.

In the Discussion, the Authors mentioned the importance of Actinobacteria in anxiety. Please, discuss how Actinobacteria changed after psychobiotic treatment in your study.

The authors describe the changes in acetic acid in the discussion section. Please, add more information about the connection between acetic acid and anxiety. 

https://www.sciencedirect.com/science/article/pii/S0002916523010572

https://www.ncbi.nlm.nih.gov/pmc/articles/PMC8622118/

Author Response

We really appreciate your review. All changes pointed out are highlighted in yellow in the manuscript.

Question 1: Firstly, a large number of typos should be corrected (for example, "citokines", "bad mental health outcomes", capitalization: "cfu or CFU" etc.)

Answer 1: We agree and all correction ere realized  

Question 2: Authors mentioned IL6 as an anti-inflammatory cytokine. However, most studies concerning psychiatric patients highlighted that this factor is engaged in the inflammatory response. Could Authors expand the information about IL6 in the Discussion section and raise the issue of an inflammatory properties?

Answer 2: We agree and expanded the discussion about IL-6

Question  3: The Introduction should be a bit shorter (compare the Introduction to the Discussion). Please, omit the least important information.

Answer 3: We agree and we reduced the introduction section 

Question 4: In the above-mentioned section Authors wrote about aggravating cognitive symptoms by antidepressants. Add information about class of drugs (SSRI, SNRI, benzodiazepines, TCA or other?)

Answer 4: We agree and added informations about class of drugs

Question 5: "However, the probiotic effect is not universal (...)" Authors should cite guidelines from The World Federation of Societies of Biological Psychiatry (WFSBP) and the Canadian Network for Mood and Anxiety Treatments (CANMAT) Taskforce. According to the WFSBP, the antidepressant effect of probiotics is not strain-dependent. Please, discuss it. https://www.tandfonline.com/doi/full/10.1080/15622975.2021.2013041

Answer  5: We added the guidelines from The World Federation of Societies of Biological Psychiatry (WFSBP) and the Canadian Network for Mood and Anxiety Treatments (CANMAT) Taskforce.

Question 6: In the Method section, the references of the scales should be added, and the information about the researcher who performed the clinical assessment.

Answer 6: The clinical assessment was performed by a nutritionist 

Question 7: In Figure 5, the Authors should edit the p-value (f.e., 4. 1 e-05???).

Answer 7: e= exponential,  This p-value is 4.1-5

Question 8: Despite no TEER (p-value) changes in point 3.4, the p-value may be close to the significance. Please, add the p-value to this section.

Answer 8: We agree and p- value was added 

Question 9: In the Discussion, the Authors mentioned the importance of Actinobacteria in anxiety. Please, discuss how Actinobacteria changed after psychobiotic treatment in your study.

Answer 9: We agree. After 14 days of probiotic treatment we observed a decreased (p<0.001) of Actinobacteria. We added this information in manuscript. 

Question 10. The authors describe the changes in acetic acid in the discussion section. Please, add more information about the connection between acetic acid and anxiety. 

https://www.sciencedirect.com/science/article/pii/S0002916523010572

https://www.ncbi.nlm.nih.gov/pmc/articles/PMC8622118/

Answer 10: We agree and added one of this paper in the manuscript